# Hierarchical Adaptive Sampling for Video Understanding

## Abstract

Due to the limited context window of Multi-modal Large Language Models (MLLMs), processing an entire video is infeasible. As a result, prevailing models mostly sample a subset of frames to substitute for the full video as input to the model. However, existing sampling methods, from simple uniform sampling to more refined methods based on query-frame relevance, employ a fixed sampling strategy that does not vary with input, failing to adapt to the diverse nature of queries, as some require comprehension of the video in its entirety, while others focus on events within short temporal segments. To address this limitation, we propose **Hierarchical Adaptive Sampling (HAS)**, a two-stage frame sampling framework. In the first stage, *Backbone Frame Construction*, we apply a Determinantal Point Process (DPP) to sample frames that are both query-relevant and non-redundant. These selected frames form the backbone of the entire sampling set, providing the foundational structure for subsequent enrichment. In the second stage, *Adaptive Contextual Enrichment*, we analyze the temporal distribution of the backbone frames to infer the query type and adaptively allocate the remaining frame budget between Local and Global Context. The Local Context enriches the backbone with fine-grained temporal dynamics and short-range causal relations, whereas the Global Context provides a holistic view of the entire video to enhance broader contextual understanding. Through a hierarchical two-stage process that integrates DPP-selected backbone frames with adaptively allocated Local and Global Context, HAS effectively addresses diverse query requirements. Incorporated into three leading MLLMs, it demonstrates consistently superior performance across the Video-MME, LongVideoBench, and MLVU benchmarks.

## 1 Introduction

Multi-modal Large Language Models (MLLMs) have demonstrated impressive performance across a wide range of image perception and reasoning tasks (Li et al., 2023; Dai et al., 2023; Liu et al., 2024a; Hurst et al., 2024; Bai et al., 2025; Comanici et al., 2025). Extending these capabilities to videos is a natural next step, yet the sheer volume of visual information in videos makes this transition far from trivial. Most current video-based MLLMs (Li et al., 2024; Shen et al., 2024; Shu et al., 2025; Zhang et al., 2025a) follow an image-based paradigm: encoding frames into visual tokens using a vision encoder, and jointly processing these tokens with text tokens in a Large Language Model. However, the constrained context window of LLMs precludes processing all video frames simultaneously; typically, a fixed subset (*e.g.*, 32 or 64) is chosen to represent the entire video. Thus, the frame sampling strategy becomes a pivotal factor influencing the model's understanding.

A common strategy is uniform sampling, which selects frames at fixed intervals. While this approach preserves the overall storyline, it can easily overlook brief yet critical moments. To address this shortcoming, relevance-based sampling, as seen in SeViLA (Yu et al., 2023) or MVU (Ranasinghe et al., 2024), effectively identifies query-relevant frames. However, this focus on relevance often causes the selection of visually similar frames and sacrifices the broader narrative context. Beneath these differences, a shared, fundamental flaw exists: they employ a fixed sampling strategy that is applied across all query types, making it unable to accommodate their diverse nature. Some queries demand a holistic understanding (e.g., "Why does the mother bird bring a fish to the fox?"), while others require a fine-grained analysis of specific events (e.g., "How did the cat manage to open the

door?"). Constrained by a fixed sampling strategy, existing methods are unable to adapt to these diverse needs, underscoring the imperative for an adaptive approach.

To overcome the limitations of fixed frame selection strategies, we propose **Hierarchical Adaptive Sampling** (HAS), a hierarchical sampling framework that adapts its frame selection to the requirements of each query. HAS operates in two stages. In the **Frame Backbone Construction** stage, we model frame sampling as a Determinantal Point Process (DPP). By incorporating a query–frame similarity term into the DPP kernel matrix, the process jointly optimizes for high relevance to the query and diversity among selected frames. This stage yields a non-redundant set of backbone frames that form a robust informational backbone for subsequent enrichment. In the **Adaptive Contextual Enrichment** stage, the remaining frame budget augments this backbone with two types of context: *local context* is captured by sampling nearby frames around each backbone frame to enhance short-term context and fine-grained temporal dynamics, while *global context* is provided via uniform sampling across the entire video timeline to preserve a comprehensive global perspective.

The adaptivity of HAS lies in how it allocates the frame budget between Local and Global Context based on the temporal distribution of backbone frames. We observe that, depending on this distribution, different allocation patterns tend to emerge. When backbone frames form a dense temporal cluster, we believe the queried object or scene is likely concentrated within a brief time window, meaning that the answer probably resides in this short segment and its immediate context. In such cases, HAS tends to assign more budget to Local Context, enriching these key moments with fine-grained temporal dynamics and short-term causalities. Conversely, when backbone frames are sparse and widely distributed, we consider that the queried subject is more likely to appear multiple times throughout the video, which often requires a comprehensive understanding of the entire timeline. HAS therefore tends to devote more budget to Global Context, connecting distant events and building a cohesive narrative. This mechanism allows HAS to flexibly shift between zooming in on specific events and constructing a holistic overview, thereby overcoming the inflexibility of static methods.

To validate the effectiveness of our method, we conducted extensive experiments by integrating HAS into three leading MLLMs: Qwen2.5-VL (Bai et al., 2025), InternVL3 (Zhu et al., 2025), and LLaVA-Video (Zhang et al., 2024). Our evaluation was performed on three widely-used benchmarks: Video-MME (Fu et al., 2025), LongVideoBench (Wu et al., 2024a), and MLVU (Zhou et al., 2024). These benchmarks collectively cover a wide spectrum of video lengths and question types, providing a robust testbed to rigorously evaluate the effectiveness and generalizability of our method. Our results demonstrate that HAS not only significantly outperforms the default uniform sampling strategy used by these models but also surpasses other advanced sampling methods, such as AKS (Tang et al., 2025).

## 2 RELATED WORK

### 2.1 MULTIMODAL LARGE LANGUAGE MODELS

The remarkable success of Large Language Models (LLMs) has motivated extending them to incorporate visual information, giving rise to Multimodal LLMs (MLLMs). Early methods employed complex vision–language alignment architectures. For example, BLIP-2 (Li et al., 2023) and Qwen-VL (Bai et al., 2023) used Q-Formers to transform visual features into the language embedding space, while Flamingo (Alayrac et al., 2022) integrated cross-attention layers directly into the LLM to fuse visual and textual representations. LLaVA (Liu et al., 2024b) was the first to show that effective visual–text alignment can also be achieved with a much simpler approach, a single Multi-Layer Perceptron (MLP), avoiding the need for extensive modifications to the base architecture. This design was later adopted by most image-based MLLMs, pushing their visual understanding capability to new heights, as exemplified by ShareGPT4V (Chen et al., 2024), MiniGPT-4V (Zhu et al., 2023), InternVL3 (Zhu et al., 2025) and Qwen2.5-VL (Wang et al., 2024a).

### 2.2 VIDEO-BASED MLLMS

Extending MLLMs from images to videos presents a key challenge: videos contain massive amounts of information, and encoding all frames into tokens for the LLM would surpass its context window capacity. Most video-based MLLMs build upon the architecture of image-based MLLMs

and address this limitation by sampling a subset of frames as input, as in Video-ChatGPT (Maaz et al., 2023), Video-LLaMA (Zhang et al., 2023), Deepseek-VL2 (Wu et al., 2024b), and Video-LLaVA (Lin et al., 2023). This strategy inevitably discards information from unsampled frames, a limitation that becomes more pronounced as the video length increases. To address this limitation, approaches such as ADACM (Man et al., 2025) and HierarQ (Azad et al., 2025) adopt transformer-based architectures to sequentially process all frames and condense them into a single compact embedding representing the entire video. However, despite mitigating the information loss introduced by frame sampling, these architectures generally underperform mainstream sampling-based MLLMs.

### 2.3 FRAME SAMPLING FOR VIDEO MLLMS

Given the limited context window of MLLMs, frame sampling has emerged as the dominant solution for adapting long videos to VideoQA tasks, making it a critical factor in overall model performance. Early influential methods such as SeViLA (Yu et al., 2023) and MVU (Ranasinghe et al., 2024) focused on query relevance by using MLLMs to assign importance scores to each frame individually. To address the efficiency bottleneck of scoring each frame individually, Hu et al. (2025) predict all frame scores in a single forward pass. Other concurrent approaches deviate from explicit query-relevance scoring at inference: Koala (Tan et al., 2024) employs fixed sparse sampling to condition subsequent visual processing, while ViLA (Wang et al., 2024b) trains an end-to-end module to learn a sampling strategy. More recent methods refine the selection criteria: AKS (Tang et al., 2025) simultaneously considers both the relevance of a frame and the temporal coverage of the already selected frames, and T* (Ye et al., 2025) converts temporal search into spatial search by arranging frames into a large grid image. However, existing methods often suffer from two limitations: they tend to produce redundant key frames, and they rely on a fixed sampling strategy that treats all queries uniformly, ignoring the diversity in evidential requirements posed by different types of questions. To address these challenges, our HAS framework introduces (1) a DPP-based frame strategy that explicitly reduces redundancy while preserving relevance, and (2) an adaptive context augmentation strategy that allocates global and local context based on the specific demands of each query.

## 3 METHOD

### 3.1 PRELIMINARIES: PROBLEM FORMULATION

To reason about our proposed method, we first formalize the video question answering pipeline. A video, $V = \{v_1, \ldots, v_T\}$, is a sequence of $T$ frames. A MLLM, $\mathcal{M}$, has a limited context window, forcing it to operate on a small subset of $K$ frames ($K \ll T$) rather than the entire video. We denote the frame sampling strategy as $\mathcal{S}$, which takes the video $V$ and optionally the query $Q$ as input and outputs a set of $K$ frame indices $\mathcal{I} = \mathcal{S}(V, Q)$. The selected frames, $\{v_i\}_{i \in \mathcal{I}}$, are then provided to the MLLM along with $Q$ to generate an answer $A$. The quality of the generated answer is therefore critically dependent on the effectiveness of the strategy $\mathcal{S}$. An ideal strategy $\mathcal{S}^*$ would always select the optimal set of indices $\mathcal{I}^*$. This optimal set is defined as the one that maximizes the probability of the MLLM generating the ground-truth answer, $A_{\text{gt}}$:

$$\mathcal{I}^* = \arg\max_{\mathcal{I} \subset \{1..T\}, |\mathcal{I}|=K} P(A = A_{\text{gt}} \mid \{v_i\}_{i \in \mathcal{I}}, Q; \mathcal{M}). \tag{1}$$

However, learning such a strategy is intractable, as even finding $\mathcal{I}^*$ for a single instance would require exhaustively evaluating all possible subsets with the MLLM. The goal of this work is therefore to design a practical and effective strategy, $\mathcal{S}_{\text{ours}}$, that serves as a proxy to $\mathcal{S}^*$.

### 3.2 HIERARCHICAL ADAPTIVE SAMPLING

Existing frame sampling methods typically adopt a fixed strategy regardless of the query type, limiting their ability to capture the most informative context for diverse VideoQA tasks. To address this limitation, we propose HAS, a two-stage frame sampling framework. In the first stage, Backbone Frame Construction, we select query-relevant yet non-redundant frames to serve as the backbone of the sampling set. In the second stage, Adaptive Contextual Enrichment, the remaining budget is adaptively allocated between Local and Global Context to enrich the backbone so as to meet the needs of different query types. The following sections detail the implementation of each stage.

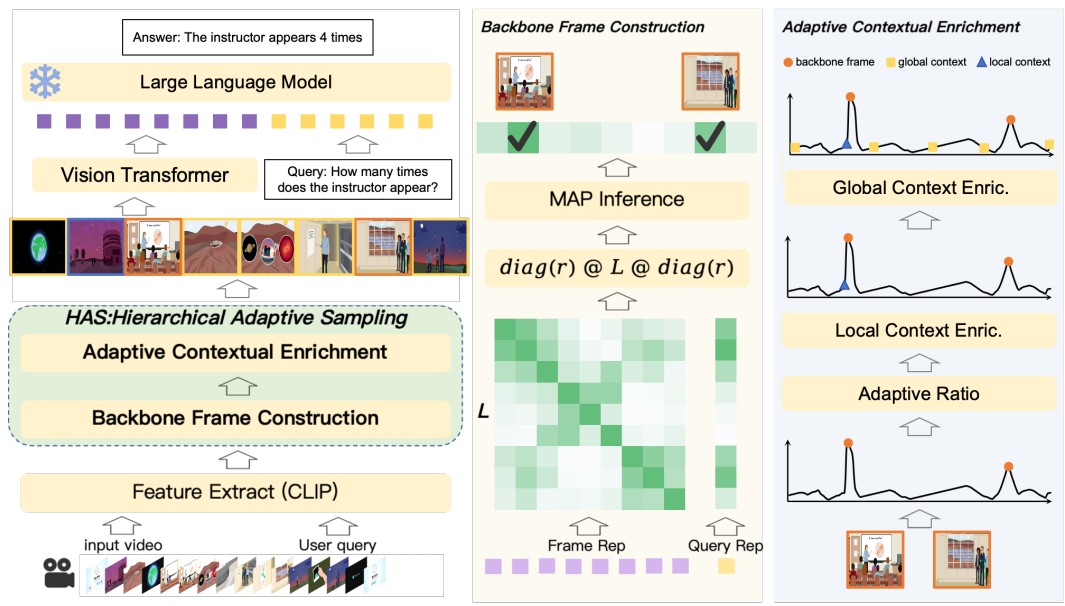

Figure 1: The overall flowchart of HAS. HAS is a training-free approach that enhances model performance solely by modifying the frames sampled from the video. In the Adaptive Contextual Enrichment module, different colors denote different frame types: orange for Backbone Frames, green for Global Context, and blue for Local Context.

### 3.2.1 BACKBONE FRAME CONSTRUCTION

HAS begins by selecting a set of backbone frames that are simultaneously highly relevant to the query and internally diverse. We model this as a subset selection problem using a Determinantal Point Process, a natural choice for balancing quality and diversity. The effectiveness of DPP has been well established in tasks like recommendation systems (Chen et al., 2018; Wilhelm et al., 2018) and, more recently, token pruning (Zhang et al., 2025b).

In our DPP formulation, we construct a kernel matrix $L$ that jointly incorporates frame-query relevance (quality) and inter-frame redundancy (diversity) to guide backbone frame selection. Specifically, we use a pre-trained CLIP model to extract embeddings for each video frame $\mathbf{f}_i$ and the query, and compute the relevance score $r_i$ as their cosine similarity. The kernel entries are given by:

$$L_{ij} = r_i \cdot \text{sim}(\mathbf{f}_i, \mathbf{f}_j) \cdot r_j, \tag{2}$$

where $\text{sim}(\cdot, \cdot)$ denotes the cosine similarity between frame embeddings.

Given $L$, DPP assigns each candidate subset $\mathcal{I}$ a probability proportional to the determinant of its corresponding submatrix, $\det(L_\mathcal{I})$, which simultaneously reflects the collective relevance of the selected frames and their diversity. Selecting the backbone frames can thus be formulated as a maximum a posteriori (MAP) inference problem under the DPP model, i.e., finding the subset of size $k_b$ that maximizes:

$$\mathcal{I}_{\text{backbone}} = \underset{\mathcal{I} \subseteq \{1..T\}, |\mathcal{I}| = k_b}{\arg\max} \det(L_\mathcal{I}), \tag{3}$$

where $k_b$ denotes the number of backbone frames to be selected.

Since this optimization is NP-hard, we employ an efficient greedy algorithm from Chen et al. (2018). This algorithm iteratively selects the sample that provides the largest marginal gain to the log-determinant until the subset size is reached. To avoid the limitations of a fixed selection size, we further introduce an adaptive stopping criterion: the process terminates early if the marginal gain falls below a predefined threshold $\eta$. This process produces the final set of backbone frame indices $\mathcal{I}_{\text{backbone}}$ for use in the subsequent stage.

### 3.2.2 Adaptive Contextual Enrichment

The backbone frames provide a strong yet sparse semantic foundation. To enrich this foundation for more complex reasoning, we utilize the remaining frame budget, $k_{\text{context}} = K - |\mathcal{I}_{\text{backbone}}|$, to sample additional context frames. This budget is adaptively allocated between two complementary types: Global Context and Local Context.

**Local Context.** While the backbone frames pinpoint critical moments, their DPP-enforced diversity often yields temporally isolated snapshots lacking the immediate context needed to capture local dynamics and short-term causalities. To remedy this, we apply a ripple expansion process to sample Local Context frames, $\mathcal{I}_{\text{LC}}$. Specifically, we initialize a queue with all backbone frames, sorted by their relevance scores, and iteratively expand until the local context budget $k_{\text{LC}}$ is reached. At each step, the frame dequeued becomes the current expansion point for a bidirectional search. Each search proceeds outward until encountering either a previously selected frame or a new frame whose similarity to the expansion point falls below a threshold $\tau$. Qualifying new frames are added to $\mathcal{I}_{\text{LC}}$ and enqueued for subsequent expansions. This process captures the local dynamics and causal relations of the backbone frames.

**Global Context.** Some queries require a holistic understanding that query-focused backbone frames alone may not provide. To address this, HAS allocates part of the frame budget to Global Context, sampling frames $\mathcal{I}_{\text{GC}}$ to provide a broad view of the video. We considered multiple GC sampling strategies during design, and preliminary evaluation indicated that simple Uniform sampling is the most effective. Therefore, Uniform sampling is adopted as the default GC method in this work, with detailed comparisons and analysis provided in Section 4.3.3.

**Adaptive Budget Allocation.** A fixed allocation between global and local context is suboptimal, as the ideal balance is query-dependent. We posit that the temporal distribution of backbone frames provides a powerful cue to the query's nature. Our core intuition is that a dense temporal cluster of backbone frames signifies a singular, localized event, which demands a fine-grained analysis. We therefore allocate more budget to Local Context to capture the local dynamics and short-term causalities. Conversely, a sparse distribution of frames implies the query pertains to an entity or event that recurs across the video. This demands a holistic understanding of the entire video; therefore, we allocate a larger portion of the budget to Global Context to connect these disparate events and build a comprehensive narrative.

To quantify this temporal distribution, we compute a scattering score, $\mathcal{S}_{\text{scatter}}$. We first augment the sorted backbone frame timestamps with the video's start (0) and end (T) points. Then, we calculate the mean ($\mu_{\text{gap}}$) and standard deviation ($\sigma_{\text{gap}}$) of the temporal gaps between these consecutive timestamps. The score is derived from their coefficient of variation:

$$\mathcal{S}_{\text{scatter}} = \exp\left(-\frac{\sigma_{\text{gap}}}{\mu_{\text{gap}} + \epsilon}\right), \tag{4}$$

where $\epsilon$ is a small stability constant. This score approaches 1 for uniformly distributed frames and 0 for highly clustered ones. And the score directly governs the allocation of the total context budget, $k_{\text{context}}$:

$$k_{\text{GC}} = \lfloor \mathcal{S}_{\text{scatter}} \cdot k_{\text{context}} \rfloor \tag{5}$$
$$k_{\text{LC}} = k_{\text{context}} - k_{\text{GC}} \tag{6}$$

This mechanism enables our strategy to dynamically trade off between a holistic overview (more global context) and a fine-grained analysis of specific events (more local context). The final set of frame indices passed to the MLLM is $\mathcal{I}_{\text{final}} = \mathcal{I}_{\text{backbone}} \cup \mathcal{I}_{\text{GC}} \cup \mathcal{I}_{\text{LC}}$.

## 4 Experiments

### 4.1 Experimental Setup

**Datasets and Models.** To empirically validate the effectiveness of our method, we conduct comprehensive experiments on three widely used video understanding benchmarks: VideoMME (Fu et al.,

Table 1: Main results on long video understanding benchmarks. The LVB-60s subset is excluded from our evaluation since our approach re-samples from an initial 1fps stream. **Bold** indicates the best performance and underline indicates the second-best within each group.

| Model | Size | Frames | LVB | | VideoMME | | | MLVU | | |
|---|---|---|---|---|---|---|---|---|---|---|
| | | | 600 | 3600 | Short | Med. | Long | Hol. | S-D | M-D |
| *Proprietary* | | | | | | | | | | |
| GPT4o | – | – | 66.7 | 61.6 | 77.7 | 66.2 | 59.8 | 79.4 | 53.7 | 35.7 |
| Gemini-1.5-Flash | – | – | 63.1 | 57.3 | 71.3 | 58.6 | 52.1 | 73.9 | 52.5 | 27.9 |
| *Open-source* | | | | | | | | | | |
| VideoChat2 | 7B | 16 | 39.0 | 37.5 | 54.2 | 42.8 | 39.9 | 71.7 | 50.0 | 24.9 |
| Long-VITA | 14B | 64 | 52.7 | 46.8 | 74.6 | 61.2 | 55.0 | 81.0 | 69.8 | 50.1 |
| LLAVA-OV | 72B | 64 | 61.6 | 56.5 | 77.2 | 63.2 | 58.2 | 83.1 | 74.9 | 48.6 |
| LLAVA-Video | 72B | 64 | 63.9 | 59.3 | 80.8 | 68.6 | 62.1 | 81.2 | 78.7 | 55.7 |
| Qwen2.5VL | 7B | 64 | 59.0 | 51.1 | 73.9 | 62.3 | 52.2 | 80.6 | 66.0 | 43.9 |
| + TOP | 7B | 64 | 62.9 | 56.0 | 74.2 | 62.3 | 53.8 | 76.5 | 72.2 | 54.4 |
| + AKS | 7B | 64 | 62.1 | 56.2 | 74.4 | 65.1 | 54.9 | **82.3** | 71.4 | 51.0 |
| + HAS | 7B | 64 | **63.8** | **56.4** | **74.8** | **65.3** | **55.4** | 80.8 | **73.4** | **57.2** |
| InternVL3 | 8B | 64 | 61.7 | 53.0 | 76.2 | 66.4 | 55.4 | **84.7** | 73.9 | 52.5 |
| + TOP | 8B | 64 | 63.6 | 56.2 | 77.8 | 63.6 | 54.6 | 80.8 | 75.4 | 57.0 |
| + AKS | 8B | 64 | 63.3 | 54.8 | 77.2 | 68.9 | 55.3 | 82.3 | 75.0 | 52.9 |
| + HAS | 8B | 64 | **64.6** | **56.4** | **78.1** | **70.0** | **56.3** | 82.3 | **77.6** | **58.1** |
| LLAVA-Video | 7B | 64 | 59.2 | 50.9 | 75.9 | 63.2 | 52.1 | 79.7 | 75.0 | 53.7 |
| + TOP | 7B | 64 | 62.1 | 53.7 | 76.9 | 59.8 | 51.9 | 75.6 | 74.4 | 57.7 |
| + AKS | 7B | 64 | 61.4 | 54.3 | 77.0 | **65.1** | **55.0** | 79.3 | 74.9 | 52.7 |
| + HAS | 7B | 64 | **63.1** | **56.2** | **77.6** | 64.3 | **55.0** | **80.6** | **76.1** | **60.6** |

2025), LongVideoBench (Wu et al., 2024a), and MLVU (Zhou et al., 2024). These benchmarks are characterized by their diverse range of question types and video lengths, with some videos extending beyond one hour. This makes them a rigorous testbed for evaluating the efficacy of frame sampling strategies. For models, we select three powerful MLLMs: Qwen2.5-VL (Bai et al., 2025), InternVL3 (Zhu et al., 2025), and LLaVA-Video (Zhang et al., 2024).

**Compared Methods.** We compare our proposed method, HAS, against three frame sampling methods. These are: (1) UNI, which samples frames at fixed temporal intervals across the video's timeline and serves as the default method for the models evaluated in this paper; (2) TOP, which selects frames in descending order of their query-relevance scores; and (3) AKS (Tang et al., 2025), an enhancement of TOP that simultaneously considers both the relevance of a frame and the temporal coverage of the already selected set of frames.

**Implementation Details.** We conduct all our experiments using the VLMEvalKit framework (Duan et al., 2024). Our method is implemented in a plug-and-play manner; we exclusively modify the input frame selection while keeping the model weights frozen throughout all evaluations. To ensure a fair and efficient comparison for all relevance-based methods (TOP, AKS, and our HAS), we adopt a common protocol used by the original AKS paper. First, we pre-sample a candidate pool from each video at 1 FPS. The respective sampling algorithms then select the final frames from this pool. Furthermore, to ensure a fair comparison, all three methods use the CLIP-L (Radford et al., 2021) to compute relevance scores.

## 4.2 MAIN RESULTS

### 4.2.1 QUANTITATIVE RESULTS

As presented in Table 1, our proposed HAS is comprehensively evaluated across three models and three benchmarks. On the great majority of evaluations, HAS outperforms the other sampling strategies, achieving the best performance.

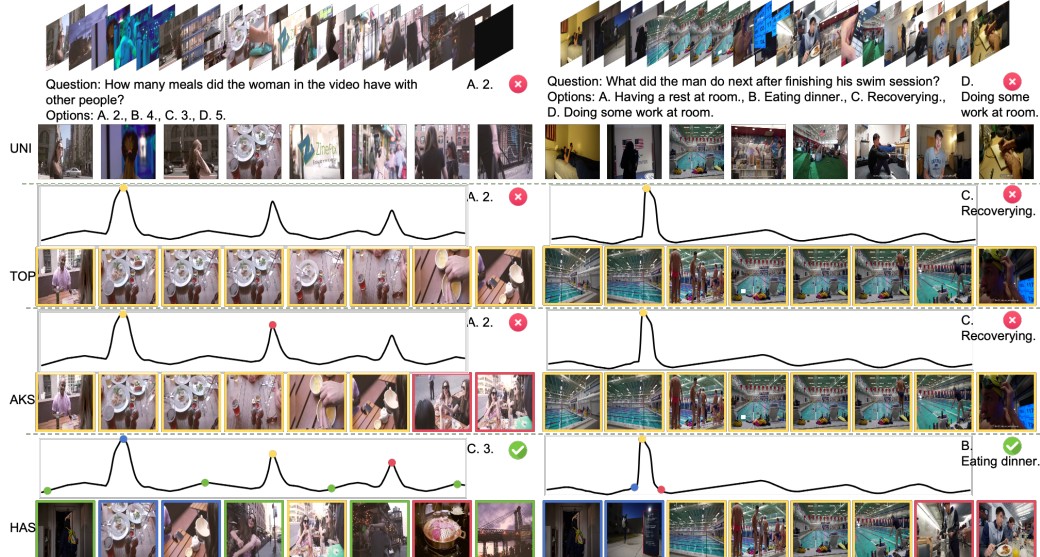

Figure 2: Two examples of how different sampling strategies impact video understanding. The curve illustrates the video's temporal relevance profile. A frame's border color matches the color of its sampling dot on the curve.

The results on the LVB and VideoMME benchmarks particularly highlight our method's advantage in long-duration video understanding. For instance, on the challenging LVB-3600 benchmark, HAS boosts the LLAVA-Video-8B model's score to 56.2, a clear +1.9 improvement over the next-best method, TOP. Similarly, on the VideoMME Long benchmark, HAS (on InternVL-3) achieves a score of 56.3, significantly outperforming AKS (55.3). Moreover, this strong performance across both LVB, which emphasizes single-scene analysis, and VideoMME, which requires integrating information from multiple moments to answer, showcases our method's ability to adapt to diverse problem types.

The results on the MLVU benchmark further showcase our method's adaptability. HAS achieves top scores on detail-oriented tasks like Single-Detail (S-D) and Multi-Detail (M-D) (e.g., 77.6 and 58.1 with InternVL3-8B), while also securing highly competitive results on Holistic (Hol.) tasks, including scores of 80.6 with LLAVA-Video and 82.3 with InternVL3, the best and second-best results among all sampling strategies. This demonstrates HAS's dual capability: providing a global perspective for holistic queries, while also capturing the local dynamics and short-term context for questions focused on specific moments.

Notably, HAS enables smaller open-source models to achieve performance competitive with, and in some cases even superior to, much larger or proprietary counterparts. For example, HAS-enhanced LLAVA-Video-7B model (76.1) surpasses the ten-times-larger LLAVA-Video-72B (74.9) on MLVU S-D, demonstrating immense gains at a fraction of the computational cost. In another compelling case, InternVL3-8B with HAS (70.0) outperforms the proprietary GPT-4o (66.2) on VideoMME Med. These results highlight a key insight: in certain scenarios, employing our economical HAS as a video sampling method is a more effective strategy for performance enhancement than the costly alternative of training ever-larger models.

### 4.2.2 QUALITATIVE RESULTS

Figure 2 illustrates the behavior of different sampling strategies on two representative cases. In the first case, HAS, benefiting from DPP, accurately captures three distinct question-relevant scenes that are all necessary for answering the question, whereas AKS and TOP ignore some of these key scenes to varying degrees. In addition, the global context provide a more complete view of how the video evolves over time, so that the final sampled frame set offers a coherent and sufficient basis for answering the question. In the second case, HAS enriches the selected query-relevant key frames with local context which supply the crucial short term context required to answer the question correctly. In comparison, AKS and TOP rely solely on key frames, which results in incomplete

Table 2: Ablation study on frame budget for HAS on InternVL3.

| Metric | VideoMME | | | LVB | | | MLVU | | |
|---|---|---|---|---|---|---|---|---|---|
| Frames | 64 | 32 | 16 | 64 | 32 | 16 | 64 | 32 | 16 |
| UNI | 66.0 | 64.4 | 61.9 | 59.8 | 56.4 | 55.1 | 73.1 | 70.0 | 65.8 |
| TOP | 65.3 | 63.5 | 61.0 | 61.8 | 60.9 | 60.0 | 74.6 | 73.0 | 69.4 |
| AKS | 67.1 | 66.2 | **65.8** | 61.1 | 59.5 | **60.1** | 74.3 | 73.1 | 65.0 |
| HAS | **68.1** | **66.7** | 64.3 | **62.2** | **61.5** | 59.2 | **75.8** | **74.1** | **70.3** |

Table 3: Ablation study of HAS components on InternVL3.

| Model | Frames | LVB | | VideoMME | | | MLVU | | |
|---|---|---|---|---|---|---|---|---|---|
| | | 600 | 3600 | Short | Med. | Long | Hol. | S-D | M-D |
| TOP | 64 | 63.6 | 56.2 | 77.8 | 63.6 | 54.6 | 80.8 | 75.4 | 57.0 |
| DPP | 64 | 63.8 | 55.7 | 78.0 | 64.8 | 55.4 | 82.2 | 76.3 | 56.8 |
| DPP+GC | 64 | 64.3 | 56.0 | 77.9 | 68.3 | **57.0** | **82.3** | 77.1 | 57.9 |
| DPP+LC | 64 | 61.7 | **56.4** | 78.0 | 67.0 | 55.2 | **82.3** | **77.9** | **59.1** |
| HAS | 64 | **64.6** | **56.4** | **78.1** | **70.0** | 56.3 | **82.3** | 77.6 | 58.1 |

contextual understanding and incorrect predictions. Jointly, these two cases collectively demonstrate the effectiveness of HAS.

## 4.3 ABLATION STUDY

### 4.3.1 PERFORMANCE UNDER DIFFERENT FRAME BUDGETS

Table 2 presents a performance comparison under varying input frame budgets (64, 32, and 16).The results demonstrate the strong performance and robustness of our method across different frame budgets. Under ample frame budgets of 64 and 32 frames, HAS consistently achieves the best performance across all benchmarks. For instance, at 32 frames, HAS secures top scores of 66.7 on VideoMME and 61.5 on LVB, clearly surpassing the next-best scores of 66.2 (from AKS) and 60.9 (from TOP), respectively. When the frame budget is constrained to 16 frames, a significant performance decline is observed for HAS. Although it still substantially outperforms UNI (e.g., 59.2 vs. 55.1 on LVB) and maintains its leading performance on MLVU (70.3) among the compared methods, it no longer surpasses AKS on the VideoMME benchmark (e.g., 64.3 vs. 65.8). We attribute this to a limitation inherent in our method's design under such extreme constraints. HAS's final frame set is composed of three distinct parts (from DPP, Local Context and Global Context). With a severely limited budget, each component receives too few frames to be effective. This results in a fragmented collection of frames that, when combined, lacks overall semantic coherence and quality.

### 4.3.2 ABLATION STUDY ON THE COMPONENTS OF HAS

To validate the contribution of each component in HAS, we conducted a detailed ablation study, with results presented in Table 3. Our ablation study (Table 3) begins by comparing DPP against TOP. DPP consistently outperforms TOP, with notable gains on benchmarks like VideoMME-Med (64.8 vs. 63.6) and MLVU S-D (76.3 vs. 75.4). This superiority stems from DPP's inherent ability to balance both relevance and diversity. Therefore, we select DPP for our backbone frame sampling.

Building on this, we observe that different context types cater to different benchmark demands. Specifically, adding Global Context (DPP+GC) yields substantial gains on benchmarks emphasizing global semantics, boosting the VideoMME-Med score from 64.8 to 68.3. Conversely, Local Context (DPP+LC) proves highly effective for tasks requiring fine-grained detail, achieving excellent scores on MLVU S-D 77.9 and M-D 59.1. These results show that GC and LC each offer unique, complementary advantages. This directly motivates our adaptive approach.

By successfully synthesizing these complementary strengths, our full method, HAS, achieves the best or second-best performance across all benchmarks. It secures top scores of 64.6 on LVB-600 and 70.0 on VideoMME-Med, and remains highly competitive even where not strictly the top performer, such as on VideoMME-Long (56.3). This robust performance validates the effectiveness of our adaptive allocation strategy and hierarchical design.

### 4.3.3 ABLATION STUDY OF ALTERNATIVE STRATEGIES FOR GLOBAL CONTEXT SAMPLING

In HAS, a portion of the budget is dedicated to Global Context to provide a global perspective of the video. To identify the most effective GC strategy, we evaluated four approaches: (i)Uniform (UNI), which samples frames at uniform temporal intervals; (ii)Max-in-Segment (MiS), which divides the video into uniform segments and selects the most relevant frame to the query from each; (iii)Clustering-based (CLUS), which performs

Table 4: Ablation of Global Context strategies.

| Method | LVB | V-MME | MLVU |
|--------|------|-------|------|
| UNI | **62.2** | **68.1** | 75.8 |
| MiS | 61.0 | 67.4 | **76.0** |
| CLUS | 61.1 | 67.9 | 75.7 |
| TBS | 61.9 | 67.4 | 75.2 |

clustering within each uniform segment and then selects the centroid of the largest cluster; and (iv)Temporal Bridging (TB), which iteratively samples the midpoint of the largest temporal gap. As shown in Table 4, the simple UNI strategy proves highly effective, with leading performance in benchmarks such as LVB and VideoMME. We attribute this to a mismatch: Global Context aims to offer a broad, unbiased temporal overview, whereas methods like MiS and CLUS introduce relevance or centrality biases. These biases conflict with this goal and compromise the uniform coverage vital for a holistic view. Therefore, we conclude that a simple unbiased strategy like Uniform sampling is the most robust choice for supplementing a query-focused backbone.

### 4.3.4 ABLATION STUDY OF HYPERPARAMETERS

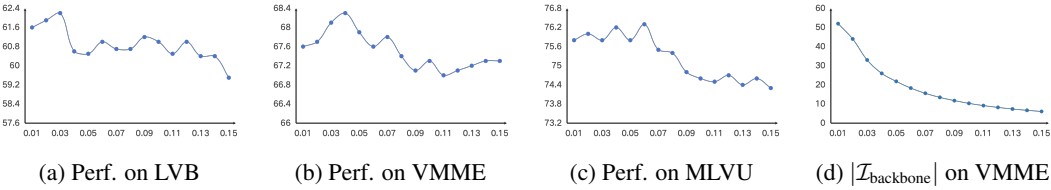

(a) Perf. on LVB      (b) Perf. on VMME      (c) Perf. on MLVU      (d) $|\mathcal{I}_{\text{backbone}}|$ on VMME

Figure 3: Analysis of the hyperparameter $\eta$. (a)-(c) show the model performance on three different benchmarks. (d) shows the number of backbone frames selected. The x-axis in all plots represents the value of $\eta$.

In this section, we analyze the hyperparameter $\eta$, the marginal gain threshold for our DPP-based backbone frame selection, where a lower value results in more frames. As shown in Figure 3, the performance exhibits a consistent trend across benchmarks, first increasing and then decreasing. When $\eta$ is low, an excessive number of backbone frames are selected. This consumes most of the budget, leaving insufficient room for the Adaptive Contextual Enrichment and thus resulting in suboptimal performance. Conversely, when $\eta$ is high, the selection becomes overly strict, yielding a backbone that is too sparse. Consequently, critical query-relevant information is lost, causing a significant performance drop. The performance peaks around $\eta = 0.03 - 0.04$, where an optimal balance is achieved. This result strongly validates the effectiveness of our two-stage design, confirming the necessity of both a robust, DPP-selected backbone and sufficient budget for subsequent contextual enrichment. Based on these findings, we set $\eta = 0.03$ for our main experiments, as it delivers the best average performance.

## 5 CONCLUSION

In this work, we observe that most video sampling methods use a fixed strategy, which makes it difficult for them to adapt to different types of queries. To address this problem, we propose HAS. HAS utilizes a hierarchical structure and an adaptive context enrichment process to create a unique set of frames for each query. For queries that require a global understanding, HAS provides a

comprehensive overview of the entire event. For queries focused on specific moments, it provides the fine-grained local dynamics and their immediate causal context. Through this adaptive approach, our method effectively handles diverse query types and consistently achieves excellent performance across multiple benchmarks and MLLMs.

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

## A APPENDIX

### A.1 THE USE OF LARGE LANGUAGE MODELS

During the preparation of this manuscript, we utilized LLMS as a writing assistance tool. Their primary role was to refine language, improve grammar, and check for spelling errors in certain passages to enhance the clarity and readability of the text.

### A.2 ADDITIONAL QUALITATIVE VISUALIZATION.

In this section, we provide a qualitative comparison of UNI, TOP, AKS, and HAS. As the figure shows, UNI often misses critical evidence frames due to its rigid sampling strategy, which forces the model to reason based on incomplete visual information. TOP, while selecting frames that are mostly directly helpful for answering the question, introduces substantial redundancy and still overlooks some frames that are only weakly related with the question but provide complementary evidence for answering it. AKS generally captures a richer set of informative moments than UNI and TOP, but its sampled frames are still less well aligned with the question than those selected by HAS. In contrast, HAS effectively discovers non-redundant, question-relevant frames distributed

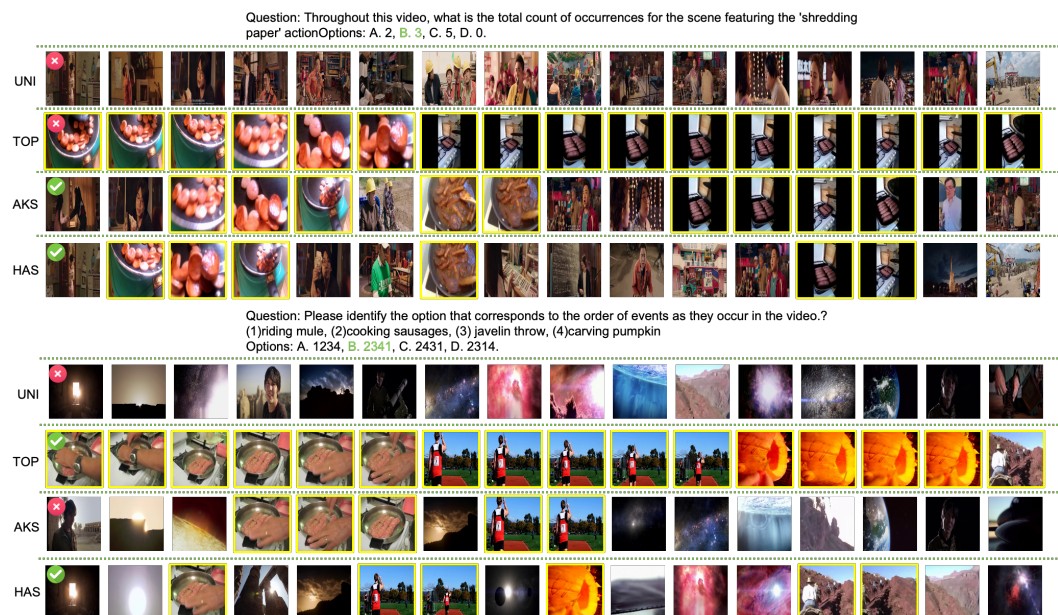

Figure 4: Two qualitative examples on questions from MLVU, comparing four frame sampling strategies: UNI, TOP, AKS, and HAS. For each example, from top to bottom, each block shows the frames sampled by UNI, TOP, AKS, and HAS, respectively.Yellow outlines highlight frames that are directly helpful for answering the question.

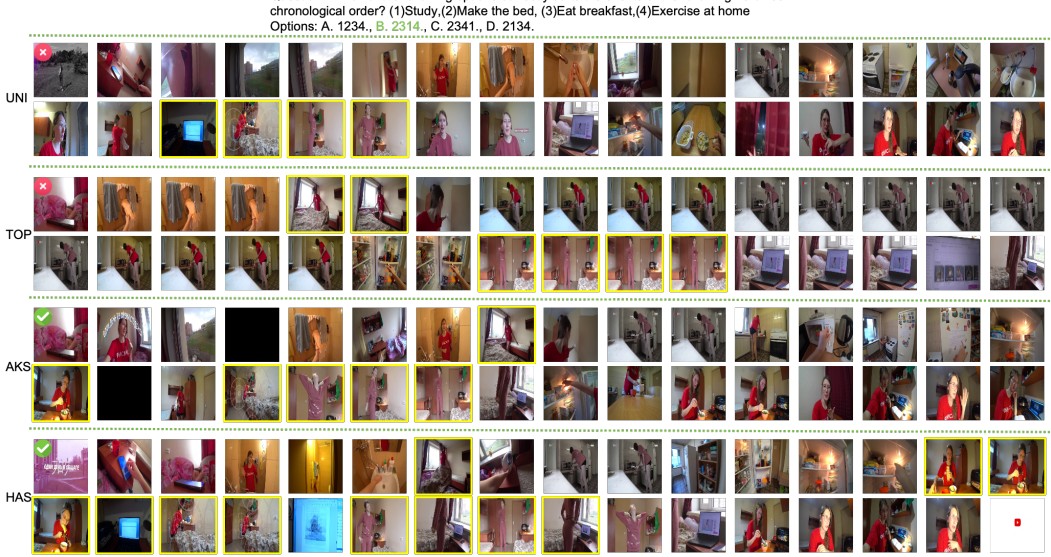

Figure 5: A qualitative result question from from VideoMME, comparing four frame sampling strategies: UNI, TOP, AKS, and HAS. For each example, from top to bottom, each block shows the frames sampled by UNI, TOP, AKS, and HAS, respectively.Yellow outlines highlight frames that are directly helpful for answering the question.

and enriches them with both local and global temporal context, thereby providing the model with a more complete and informative visual basis for reliable answer prediction.

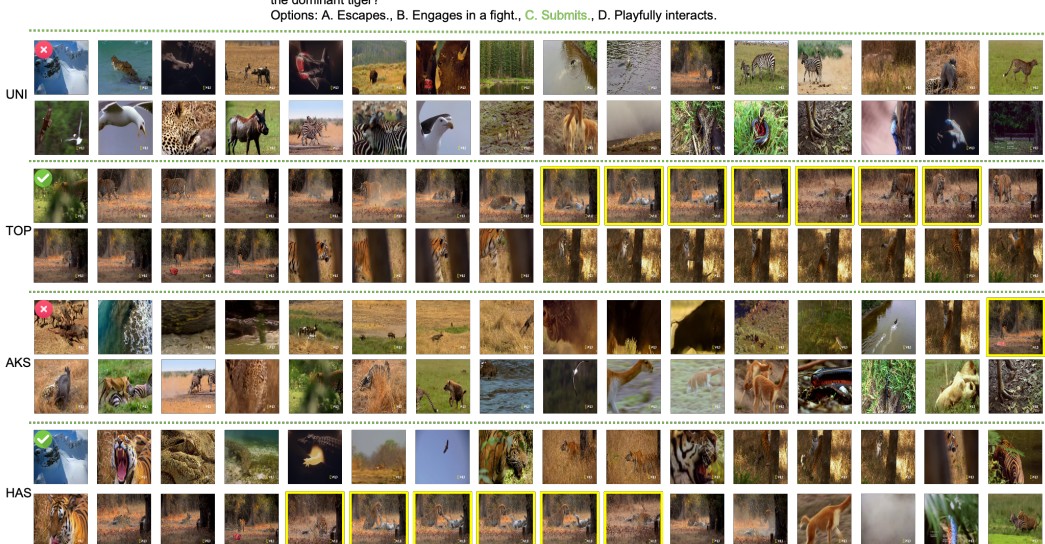

Figure 6: A qualitative result question from from VideoMME, comparing four frame sampling strategies: UNI, TOP, AKS, and HAS. For each example, from top to bottom, each block shows the frames sampled by UNI, TOP, AKS, and HAS, respectively.Yellow outlines highlight frames that are directly helpful for answering the question.

Table 5: Ablation study of different vision encoders for HAS. We report accuracy (%) on two long-video benchmarks.

| Vision encoder | VideoMME | LongVideoBench |
|---|---|---|
| CLIP | 68.1 | 62.2 |
| SigLIP2 | 68.3 | 62.5 |
| Δ (SigLIP2 − CLIP) | +0.2 | +0.3 |

### A.3    ABLATION STUDY OF DIFFERENT VISUAL ENCODERS

To quantify the impact of the underlying vision encoder on HAS, we conduct an ablation study comparing CLIP and SigLIP2 on videomme and longvideobench. As shown in Table 5, although SigLIP2 is substantially stronger than CLIP at the image–text level, the resulting performance gains on these benchmarks are quite small.

Figure 7: A qualitative result question from from MLVU, comparing four frame sampling strategies: UNI, TOP, AKS, and HAS. For each example, from top to bottom, each block shows the frames sampled by UNI, TOP, AKS, and HAS, respectively. Yellow outlines highlight frames that are directly helpful for answering the question.

Question: In which order are the following steps introduced in this video?(a) Removing a washing machine door seal.(b) Astonishing Results of the bleach on the washing machine door seal.(c) Washing machine hoses to clean to remove Mold Mildew fungus.(d) Cleaning mold out of soap draw.
Options: A. (b)(c)(a)(d)., B. (a)(b)(c)(d)., C. (a)(c)(b)(d)., D. (b)(d)(a)(c).

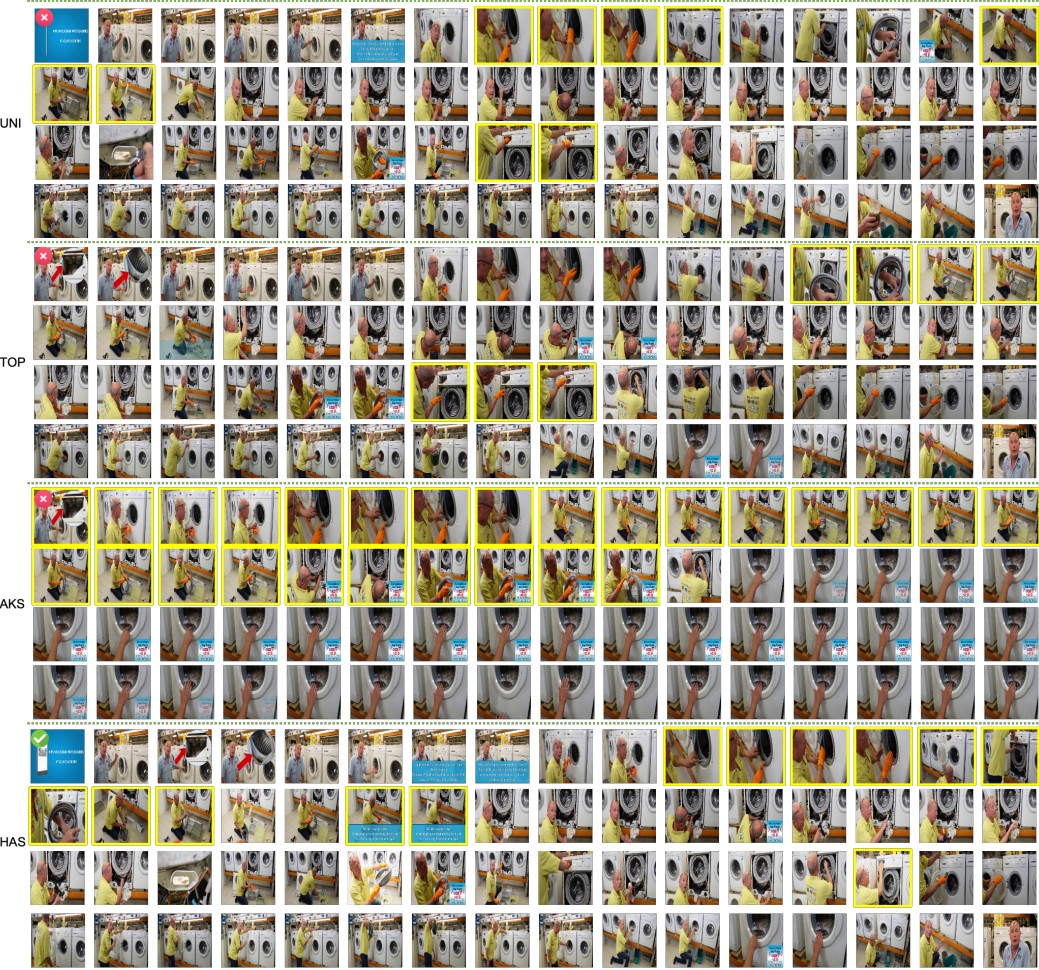

Figure 8: A qualitative result question from VideoMME, comparing four frame sampling strategies: UNI, TOP, AKS, and HAS. For each example, from top to bottom, each block shows the frames sampled by UNI, TOP, AKS, and HAS, respectively.Yellow outlines highlight frames that are directly helpful for answering the question.

