# OpenReview forum: "HIERARCHICAL ADAPTIVE SAMPLING FOR VIDEO UNDERSTANDING"
_ICLR.cc/2026/Conference — Submitted to ICLR 2026_

### Official Review · Reviewer_CRZf · 2025-10-30

**Soundness:** 3
**Presentation:** 3
**Contribution:** 3
**Rating:** 6
**Confidence:** 5

**Summary:**

The paper introduces Hierarchical Adaptive Sampling (HAS), a two-stage frame sampling framework designed to capture both local and global video content. Starting from a predefined frame budget, HAS dynamically allocates sampling, focusing on global information when the video content is sparse and emphasizing local details when temporal density is high. The experiments are comprehensive and well-explained, demonstrating the effectiveness and adaptability of HAS.

**Strengths:**

- The method is simple yet impactful. The components of HAS are well-designed and grounded in solid intuition. It effectively considers both local and global video contexts while remaining efficient and flexible.
- HAS consistently outperforms prior frame-sampling methods across three benchmarks.
- The ablation studies are thorough and strongly support the effectiveness of the proposed method.

**Weaknesses:**

While the paper is generally strong, there are a few aspects that could benefit from additional clarification or improvement:
- In Section 2.3, I recommend that the authors explicitly clarify which aspects are overlooked in previous frame-sampling methods and how HAS addresses or improves upon these limitations. The current version primarily lists prior methods without clearly contrasting them or highlighting HAS’s specific advancements.
- Have authors investigated the effect of the threshold $\tau$ in Adaptive Contextual Enrichment? Additionally, it would be great if authors can improve the part of Adaptive Contextual Enrichment in Fig 1.
- The evaluation is comprehensive; but it mainly considers video question answering benchmarks. It would be interesting to examine whether the method can also improve other video understanding capabilities, such as temporal grounding [1, 2] and grounded video question answering [3].

**References**

[1] ReXTime: A Benchmark Suite for Reasoning-Across-Time in Videos, NeurIPS 2024

[2] On the Consistency of Video Large Language Models in Temporal Comprehension, CVPR 2025

[3] Can I Trust Your Answer? Visually Grounded Video Question Answering, CVPR 2024

**Questions:**

See the Weaknesses

---

> ### Author Response · Authors · 2025-11-23
>
> ### Q1. In Section 2.3, I recommend that the authors explicitly clarify which aspects are overlooked in previous frame-sampling methods and how HAS addresses or improves upon these limitations. The current version primarily lists prior methods without clearly contrasting them or highlighting HAS’s specific advancements.
>
> **Response:**
> Thank you for constructive feedback!
>
> Following your comment, we have revised Section 2.3 to explicitly clarify the limitations overlooked in existing frame-sampling approaches and how HAS addresses these issues.
>
> ---
>
> ### Q2. Have authors investigated the effect of the threshold $\tau$ in Adaptive Contextual Enrichment? Additionally, it would be great if authors can improve the part of Adaptive Contextual Enrichment in Fig 1.
>
> **Response:**
> Thanks!
>
> We have investigated the sensitivity of the threshold $\tau$ used in Adaptive Contextual Enrichment. Our experiments show that HAS is largely insensitive to this hyperparameter when the threshold is relatively high: for $\tau$ in the range 0.6–0.9, the performance on VideoMME remains stable with only minor fluctuations. In contrast, when $\tau$ is reduced below 0.5 (i.e., $\tau \in [0,0.5]$), we observe that the performance decreases as $\tau$ becomes smaller, with a relatively large overall drop compared to the high-threshold regime.
>
> | $\tau$ |  **0.9** | **0.8**| **0.7**| **0.6** | **0.5**| **0.4**| **0.3** | **0.2**| **0.1**|
> |------------|---------|---------|---------|---------|---------|---------|---------|---------|---------|
> |  Acc |   68.0     |  68.2 |  68.1      | 67.7  | 67.9 |   67.5     |  67.1  | 66.5 | 66.7
>
> Following your suggestion, we have also improved the visualization of the Adaptive Contextual Enrichment module in Fig. 1.
>
> ---
>
> ### Q3. The evaluation is comprehensive; but it mainly considers video question answering benchmarks. It would be interesting to examine whether the method can also improve other video understanding capabilities, such as temporal grounding and grounded video question answering?
>
> **Response:**
> Thank you for constructive feedback!
>
> We have evaluated HAS on RexTime. On this benchmark, HAS consistently improves over uniform sampling (UNI). We believe these gains arise because HAS selects frames that are more semantically aligned with the query, enabling the model not only to answer more accurately but also to better localize the visual evidence that supports the answer.
>
>
> | **ReXTime** |  mIoU  | Iou>0.5 |  Acc  | Acc@IoU>0.5 |
> |:-----------:|:------:|:-------:|:----:|:-----------:|
> |     UNI     |  30.2  |  28.0   | 78.0 |    21.2     |
> |     HAS     |  31.9  |  30.1   | 80.3 |    23.7     |

---

> > ### Comment · Reviewer_CRZf · 2025-11-25
> > **Reply to the authors**
> >
> > I appreciate the authors' effort in addressing my concerns.
> >
> > However, several points still require clarification:
> > - I had difficulty fully understanding the results presented in response to Q2. From my understanding, an overly high $\tau$ would emphasize global content while overlooking local frame details, potentially leading to suboptimal performance. **Yet, the reported results show minimal differences within the range (0.6–0.9).** Are the backbone frames by DDP already sufficiently strong to compensate for this effect? Could the authors elaborate on this point?
> >
> > - Please clarify which backbone model was used in the results for Q2 and Q3 for better understanding. (+ reported metrics in the results for Q2)

---

> > > ### Author Response · Authors · 2025-11-25
> > >
> > > **Q1. I had difficulty fully understanding the results presented in response to Q2. From my understanding, an overly high
> > >  would emphasize global content while overlooking local frame details, potentially leading to suboptimal performance. Yet, the reported results show minimal differences within the range (0.6–0.9). Are the backbone frames by DDP already sufficiently strong to compensate for this effect? Could the authors elaborate on this point?**
> > >
> > > Thank you for the question and for pointing out the potential confusion in our description.
> > >
> > > In our method, $\tau$ is used to control the maximum similarity for selecting local context frames around each backbone frame. Concretely, starting from a backbone frame as the expansion point, we perform a bidirectional search over neighboring frames. The search continues outward until we encounter a frame whose similarity to the current expansion point falls **below** $\tau$; this frame is then added to the local context and becomes a new expansion point.
> > >
> > > Because of this design, the effect of $\tau$ is opposite to the intuition in the question:
> > >
> > > - A larger $\tau$ makes the condition “similarity < $\tau$” easier to satisfy, so the search stops earlier and the selected local context frames stay **closer** to the backbone frame, preserving the intended local dynamics.
> > > - A smaller $\tau$ forces the search to move **further** away until the similarity drops below this smaller threshold, so the selected “local” frames can drift to more distant content.
> > >
> > > When $\tau$ is set too low, the selected local context frames become too distant from the backbone frames, which deviates from our original intention of modeling local dynamics and thus leads to performance degradation.
> > >
> > >
> > > ---
> > >
> > > **Q2. Please clarify which backbone model was used in the results for Q2 and Q3 for better understanding.**
> > >
> > > For Q2, all reported results are based on **InternVL3-8B** as the backbone, using the same setup as in our main experiments.
> > >
> > > For Q3, we instead use **GPT-4o** as the backbone, because InternVL3-8B exhibits generally weak grounding behavior in our preliminary trials, making it difficult to fairly assess the contribution of our frame selection.

---

> > > > ### Comment · Reviewer_CRZf · 2025-11-25
> > > > **Reply to the authors**
> > > >
> > > > Thanks for the authors' detailed responses. I have no further questions and will keep my positive score as is.

---

### Official Review · Reviewer_Hx4z · 2025-10-30

**Soundness:** 3
**Presentation:** 3
**Contribution:** 2
**Rating:** 4
**Confidence:** 4

**Summary:**

This paper addresses a critical and timely challenge in Multi-modal Large Language Models (MLLMs): the limited context window makes processing long, full videos computationally infeasible. Authors propose propose Hierarchical Adaptive Sampling (HAS) consists of Backbone Frame Construction, and Adaptive Contextual Enrichment for adaptively allocate the remaining frame budget between Local and Global Context.

**Strengths:**

+ Complementary sampling strategies were proposed to jointly consider key frames, neighboring context and global context.

+ Good results are achieved, outperforming unform sampling.

+ The paper is well written.

**Weaknesses:**

1) The benefit of using DPP over TOP is not significant, being completive.  This alleviated the contribution of this paper.

2) How to allocate the budge for DPP and others and their influence on performance should be discussed and analyzed. Is the percentage of DPP frames adaptive to videos? What are the statistics of the percentages?

3) Around (4), is there typo? In my understanding, for the highly clustered ones, the standard deviation is small, the score should approach 1 instead of 0. k_{GC} should be small. Does (4) have typo?

4) The computation complexity could be analyzed by comparing different baselines.

**Questions:**

See weakness part.

---

> ### Author Response · Authors · 2025-11-23
>
> ### Q1. The benefit of using DPP over TOP is not significant, being completive. This alleviated the contribution of this paper.
>
> **Response:**
> Thanks！
>
> Regarding the benefit of using DPP over TOP, we would like to clarify that DPP brings meaningful and consistent improvements on benchmarks that require broad semantic coverage. For instance, on VideoMME-medium, our DPP-based backbone improves accuracy from 63.6 → 64.8, and on MLVU-Holistic, it improves from 80.8 → 82.2. These benchmarks emphasize long-range semantic reasoning rather than relying on a single highly relevant frame. In such settings, DPP’s ability to select non-redundant and globally diverse keyframes allows the model to access complementary visual evidence from different temporal segments, leading to stronger overall performance. In contrast, TOP tends to select the frames that are most similar to the query and therefore concentrates heavily on a few highly discriminative moments. This behavior makes TOP effective on benchmarks such as LongVideoBench, where many questions can be answered from a single frame.
>
> These empirical observations directly motivate our hierarchical sampling strategy in HAS. While DPP is effective at capturing representative keyframes that are both diverse and relevant to the query, its non-redundant nature introduces two key limitations: (1) it tends to under-capture local dynamics and short-term context, making it less sensitive to fine-grained temporal continuity or causal cues; and (2) by focusing primarily on query-relevant keyframes, it overlook frames that describe the global temporal evolution of the video but are only weakly correlated with the query. Our hierarchical design explicitly compensates for these weaknesses: Local Context Enrichment augments each DPP-selected keyframe with neighboring frames to recover local dynamics and short term context, while Global Context Enrichment introduces frames that track the overall evolution of the video, providing a complementary global perspective on how the scene changes over time. As confirmed by our ablation studies, HAS consistently outperforms DPP and maintains strong performance across heterogeneous benchmarks.
>
>
> ---
>
> ### Q2. How to allocate the budge for DPP and others and their influence on performance should be discussed and analyzed. Is the percentage of DPP frames adaptive to videos? What are the statistics of the percentages? What are the statistics of the percentages?
>
> **Response:**
> Thanks！
>
> In our implementation, the DPP backbone is obtained via a greedy selection procedure: at each step, we add the frame that yields the largest marginal gain, jointly accounting for its semantic relevance to the query and its non-redundancy with the already selected frames. The DPP selection process terminates once the maximum marginal gain over all remaining frames falls below a predefined threshold η, and the remaining frame budget is used in the Adaptive Contextual Enrichment stage.
>
> Fig. 3 illustrates the impact of different η on model performance and the number of DPP-selected frames. Fig. 3(d) shows the number of frames selected by DPP under η, while Fig. 3(a–c) reports the corresponding model performance. The results indicate that the best performance occurs when DPP-selected frames make up about half of the total budget. Allocating too many or too few frames to DPP results in a performance drop.
>
> The percentage of DPP-selected frames is adaptive to each video. Since DPP selects diverse frames that are relevant to the query, the proportion naturally varies with the video content and query. When only a few frames are relevant to the question, DPP selects fewer frames; conversely, when the video contains many non-redundant frames related to the query, the number of DPP-selected frames increases accordingly. The table below reports, under η of 0.03 on the VideoMME dataset, the number of samples whose DPP-selected frame counts fall into each interval.
>
> | Number of frames selected by DPP |  1-8  |  9–16 | 17–24 | 25–32 | 33–40 | 41–48 | 49–56 | 57–64 |
> |:----:|:---:|:-----:|:-----:|:-----:|:-----:|:-----:|:-----:|:-----:|
> |  **Number of samples**  | 136 |  405  |  512  |  520  |  388  |  264  |  180  |  295  |

---

> ### Author Response · Authors · 2025-11-23
>
> ### Q3. Around (4), is there typo? In my understanding, for the highly clustered ones, the standard deviation is small, the score should approach 1 instead of 0. k_{GC} should be small. Does (4) have typo?
>
> **Response:**
> Thanks！
>
> We would like to clarify that there is no typo. The key point is that the variance is calculated not over the raw frame timestamps, but over the temporal gaps between consecutive backbone frames, including the augmented start (0) and end (T) frames. For a uniformly distributed set of n backbone frames, the augmented frames result in n+1 temporal gaps of equal length, leading to a standard deviation of exactly zero. In contrast, when the backbone frames are temporally clustered, the temporal gaps vary significantly, resulting in a larger standard deviation. This causes the coefficient of variation, and thus K_GC, to decrease toward zero.
>
> ---
>
> ### Q4. The computation complexity could be analyzed by comparing different baselines.
>
> **Response:**
> Thank you for constructive feedback!
>
> We evaluated the average time required by different methods to perform key frame selection on the videomme benchmark. The results show that our method requires a similar amount of time as AKS and TOP, while being substantially faster than T* and VideoTree.
>
> TOP, AKS, and HAS exhibit similar runtime because all three methods first use a CLIP model to compute the similarity between frame and query, and then apply different frame selection strategies based on these similarity scores. In our implementation, for example, the total 5.46 s + 43 ms required by HAS consists of 5.46 s spent on computing frame–query similarities and only 43 ms on the subsequent frame selection. Since the similarity computation dominates the overall runtime, the additional complexity of the HAS selection strategy contributes only a negligible overhead, resulting in a total time cost that remains comparable to that of AKS and TOP.
>
> | **method** | **T*** | **VideoTree** | **TOP** | **AKS** | **HAS** |
> |:------------:|:------:|:-------------:|:------:|:------:|:------:|
> |  time    | 101.24s |    12.64s     | 5.46s+1ms | 5.46s+2ms | 5.46s+43m |

---

> > ### Comment · Reviewer_Hx4z · 2025-11-26
> >
> > Most of my concerns have been addressed. For the local context, leveraging a threshold $\tau$ actually introduces redundancy, since a couple of very similar frames will be introduced. Why not using manner as introduced in (3) to unified to increase the number? What are the disadvantages if you do so?

---

> > > ### Author Response · Authors · 2025-11-26
> > >
> > > ### Q1. Most of my concerns have been addressed. For the local context, leveraging a threshold actually introduces redundancy, since a couple of very similar frames will be introduced. Why not using manner as introduced in (3) to unified to increase the number? What are the disadvantages if you do so?
> > >
> > > **Response:**
> > > Thanks!
> > >
> > > Instead of unifying frame selection using the mechanism in (3), we deliberately design the local context as a complementary component to DPP. The DPP module selects frames based on both question relevance and non-redundancy. However, relying solely on DPP-based selection has two drawbacks: (1) the selected frames tend to be temporally isolated, which weakens the modeling of local dynamics; and (2) since DPP explicitly favors frames that are highly relevant to the question, it tends to discard short-term contextual frames around key moments that are not directly question-related but are still crucial for reasoning. For example, for the question “What did the person do before swimming?”, frames immediately preceding the swimming action are important, even if they are not themselves highly relevant under the DPP scoring.
> > >
> > > We introduce local context precisely to address these two issues. By expanding each DPP-selected frame with its local temporal neighborhood, we intentionally allow a certain amount of redundancy in order to strengthen local motion cues and preserve short-term context. At the same time, we control this redundancy by using a similarity threshold τ to avoid selecting frames that are almost identical to the backbone frames, such as near-duplicate shots in near-duplicate shots in studio news or talk-show settings.
> > >
> > > Empirically, our ablation studies support this design: under the same frame budget, “DPP + Local Context” outperforms using DPP alone on many benchmarks. Moreover, the second case in Fig. 2 and the second case in Fig. 4 qualitatively demonstrate how local context helps capture short-term temporal dependencies.

---

> > > > ### Comment · Reviewer_Hx4z · 2025-11-26
> > > >
> > > > Thanks for the efforts! I raised the score.

---

### Official Review · Reviewer_6Z8Q · 2025-10-31

**Soundness:** 3
**Presentation:** 3
**Contribution:** 3
**Rating:** 4
**Confidence:** 4

**Summary:**

This paper introduces Hierarchical Adaptive Sampling (HAS) to improve video understanding in multi-modal large language models. HAS first selects a diverse, query-relevant backbone of frames and then adaptively enriches local or global context based on their temporal distribution. Experiments on multiple benchmarks show HAS outperforms uniform and existing advanced sampling methods, effectively balancing fine-grained and holistic video comprehension.

**Strengths:**

The integration of Determinantal Point Processes (DPP) with adaptive frame selection is an interesting and well-motivated idea. The paper is also clearly written and easy to follow.

**Weaknesses:**

The proposed hierarchical frame sampling strategy shows several overlaps with existing approaches and raises key concerns:

Similarity to prior works: The first stage, query-aware frame selection, closely resembles HierarQ (Azad et al., CVPR 2025), while the second stage, which divides frames into local and global contexts, follows ideas similar to HierarQ and VideoTree (Wang et al., CVPR 2025).

Non-redundant frame selection: The selection of diverse frames is not novel - many prior long-video methods employ alternatives such as cosine-similarity merging (HierarQ, MovieChat, MA-LLM), token merging (ToMe), or adaptive clustering (VideoTree).
The proposed NP-hard probabilistic optimization seems unnecessarily complex, and the paper does not provide comparisons with these existing, often simpler, strategies. Additionally, there is no discussion of the computational overhead introduced by multi-stage, multi-scale sampling.

Over-aggressive sampling: The pipeline first subsamples videos at 1 FPS, then further reduces them to 64 frames, which is already standard input for most models. This raises doubts about scalability to truly long videos (e.g., MovieChat-1K, which can exceed 12 000 frames) — how would the model handle such scenarios when ultimately limited to just 64 frames?

Failure on dynamic datasets: On datasets with frequent scene or viewpoint changes (e.g., LiveSportsQA, Ego4D, YouCook2), such aggressive frame reduction will likely miss key temporal information and fail to capture fast-changing events. This limitation becomes even more critical for UCF-Crime or temporal grounding tasks like Charades-STA, where temporal continuity is essential.

Dataset bias: As noted in Table 2 of the NeurIPS 2024 "MMBench-Video: A Long-Form Multi-Shot Benchmark for Holistic Video Understanding" benchmark paper, VideoMME achieves around 54% accuracy using only a single frame, suggesting it rewards static visual perception rather than true long-term reasoning. The success of HAS may therefore stem from the frame-level nature of the benchmark, rather than genuine temporal understanding.

Supporting evidence: Prior work such as "Revealing Single Frame Bias for Video-and-Language Learning" has shown that single-frame cues can often suffice for many VideoQA tasks. Thus, while HAS performs well on current benchmarks, it may not reflect real long-term temporal reasoning performance.

Inconsistency in performance improvements: In several of the ablations, it seems like HAS is not always helping. It is sometimes hurting the performance.

**Questions:**

Relation to prior work: Your hierarchical frame sampling strategy shows similarities to methods like HierarQ and VideoTree. Could you clarify what distinct contributions HAS makes beyond these prior approaches, particularly in terms of query-aware selection and local/global context modeling?

Scalability and efficiency: Given that HAS first subsamples videos at 1 FPS and then selects only 64 frames, how does the method scale to very long videos (e.g., >10,000 frames) or datasets with frequent scene changes? Additionally, can you provide details on the computational overhead of the multi-stage, probabilistic sampling compared to simpler alternatives like token merging or adaptive clustering?

Temporal reasoning vs. benchmark bias: Many benchmarks used (e.g., VideoMME) show strong performance with even a single frame, raising the possibility that HAS may leverage frame-level cues rather than true long-term temporal reasoning. How does HAS perform on datasets or tasks requiring fine-grained temporal continuity, and can you provide evidence that its gains reflect genuine temporal understanding rather than dataset bias?

---

> ### Author Response · Authors · 2025-11-23
>
> ### Q1. Relation to prior work: Your hierarchical frame sampling strategy shows similarities to methods like HierarQ and VideoTree. Could you clarify what distinct contributions HAS makes beyond these prior approaches, particularly in terms of query-aware selection and local/global context modeling?
>
> **Response:**
> Thanks!
>
> Although HAS and HierarQ both aim to improve long-video understanding, they tackle fundamentally different problems and are built on entirely different technical principles. HierarQ introduces a new model architecture that sequentially processes all video frames via two separate pathways to handle short-range and long-range temporal dependencies. Its objective is to construct a more expressive video representation through autoregressive modeling.
>
> By contrast, HAS is a completely training-free, model-agnostic sampling strategy that can be plugged into any existing Video-MLLM without modifying its architecture. Our focus is on deciding which frames should be fed into the model under strict frame-budget constraints, rather than on designing a new temporal understanding module. Consequently, HAS is fundamentally a frame selection method, not a video representation learning approach. The notions of “local” and “global” in the two methods also differ substantially. In HAS, local and global frames are explicitly sampled around DPP-selected keyframes and across the full timeline, respectively. In HierarQ, short-range and long-range context are defined only along its autoregressive encoding order and internal memory.
>
> We also clarify that our method is substantially different from VIDEOTREE. First, the goals are distinct. VIDEOTREE proposes a complete training-free LVQA framework that converts a long video into a hierarchical tree of captions and lets a text-only LLM reason purely over this textual representation. In contrast, our method does not design a new QA system; instead, we specifically focus on the frame sampling problem under a fixed token budget for existing video MLLMs. HAS is a plug-and-play sampling module that can be inserted into arbitrary video MLLMs without any retraining.
>
> Second, although both methods adopt a multi-level or “hierarchical” design, the meaning of this hierarchy and the resulting behavior are fundamentally different. At the first level, HAS formulates backbone selection as a DPP-based subset selection problem that explicitly extracts frames that are both query-relevant and non-redundant, whereas VIDEOTREE relies on clustering to partition the video into groups of semantically similar events. AAt the second level, VIDEOTREE further refines each cluster by breaking it down into finer-grained nodes, capturing more query-relevant visual details. In contrast, HAS takes a completely different route: it explicitly considers the nature of the question and adaptively allocates the remaining frame budget between global context and local context. This question-aware allocation leads to a more flexible sampling policy that can better adapt to queries with different dependencies on global narrative semantics versus local, short-term details.
>
> ---
>
> ### Q2. Scalability and efficiency: Given that HAS first subsamples videos at 1 FPS and then selects only 64 frames, how does the method scale to very long videos (e.g., >10,000 frames) or datasets with frequent scene changes?
>
> **Response:**
> Thank you for constructive feedback!
>
> By default, we adopt a 1 FPS pre-sampling rate as a practical configuration, primarily to avoid excessive computational overhead when calculating frame–query similarity; however, this choice is not inherent to our method, and for shorter or more information-dense videos, a higher FPS can equally be used. Regarding the scalability of our method to very long videos, we choose a budget of 64 frames primarily because this is the default setting adopted by several mainstream video understanding models, and aligning with this configuration ensures a fair and consistent comparison. Importantly, HAS itself is not tied to a fixed number of frames: it can be instantiated with arbitrary frame budgets, and thus naturally extends to settings with more frames when additional computational resources are available, enabling it to scale to longer videos and diverse application requirements.
>
> We also evaluated our method on the MovieChat-1k benchmark, where the results show that our frame sampling approach yields a noticeable improvement in performance compared with uniform sampling, suggesting that our method is capable of handling longer videos. Moreover, on the long subset of Video-MME and the 3600 subset of LongVideoBench, our method achieves substantial gains over uniform sampling, further demonstrating its effectiveness in enhancing model understanding of long videos.
>
> | **benchmark** |  **UNI** | **HAS**|
> |------------|---------|---------|
> | moviechat        |    40.5    |  42.9 |

---

> ### Author Response · Authors · 2025-11-23
>
> ### Q3. Additionally, can you provide details on the computational overhead of the multi-stage, probabilistic sampling compared to simpler alternatives like token merging or adaptive clustering?
>
> **Response:**
> Thanks！
>
> We evaluated the average time required by different methods to perform key frame selection on the videomme benchmark. The results show that our method requires a similar amount of time as AKS and TOP, while being substantially faster than T* and VideoTree.
>
> TOP, AKS, and HAS exhibit similar runtime because all three methods first use a CLIP model to compute the similarity between frame and query, and then apply different frame selection strategies based on these similarity scores. In our implementation, for example, the total 5.46 s + 43 ms required by HAS consists of 5.46 s spent on computing frame–query similarities and only 43 ms on the subsequent frame selection. Since the similarity computation dominates the overall runtime, the additional complexity of the HAS selection strategy contributes only a negligible overhead, resulting in a total time cost that remains comparable to that of AKS and TOP.
>
> | **method** | **T*** | **VideoTree** | **TOP** | **AKS** | **HAS** |
> |:------------:|:------:|:-------------:|:------:|:------:|:------:|
> |  time    | 101.24s |    12.64s     | 5.46s+1ms | 5.46s+2ms | 5.46s+43m |
>
> We did not compare token merging because its goal is to reduce computational costs while maintaining performance. In contrast, our frame sampling approach aims to enhance performance by only modifying the input. Given these differing objectives, token merging is not well-suited for our task.
>
> ---
>
> ### Q4. The success of HAS may therefore stem from the frame-level nature of the benchmark, rather than genuine temporal understanding.
>
> **Response:**
> Thank you for constructive feedback!
>
> MMbench points out that some benchmarks can be answered with just a single frame. Based on this observation, it proposes the mmbench benchmark, which requires integrating information from different time segments and understanding the entire video to provide correct answers. Therefore, we conducted experiments on mmbench, and the results showed a significant improvement, with HAS outperforming UNI by a large margin, from 1.7 to 1.96.
>
> | **benchmark** |  **UNI** | **HAS**|
> |------------|---------|---------|
> | MMbench         |   1.7     | 1.96 |
>
> ---
>
> ### Q5. How does HAS perform on datasets or tasks requiring fine-grained temporal continuity, and can you provide evidence that its gains reflect genuine temporal understanding rather than dataset bias?.
>
> **Response:**
> Thank you for constructive feedback!
>
> To specifically assess how HAS behaves on tasks that require fine-grained temporal continuity, we evaluated our method on the EgoSchema subset, which is designed to test detailed temporal reasoning over long egocentric videos. HAS achieves a notable improvement over the uniform sampling.
>
> | **benchmark** |  **UNI** | **HAS**|
> |------------|---------|---------|
> | Ego-scheme (subset)     |   71.4    | 74.4 |
> ---
>
> ### Q6. Inconsistency in performance improvements: In several of the ablations, it seems like HAS is not always helping. It is sometimes hurting the performance.
>
> **Response:**
> Thanks！
>
> We acknowledge that HAS does not achieve the absolute best score in every single ablation; however, it consistently delivers either the best or second-best performance across all benchmarks. When HAS is not the top-performing variant, the best alternative is almost always DPP+GC or DPP+LC. This is expected because fixed variants (GC-only or LC-only) are inherently biased toward a particular type of question. For tasks requiring predominantly holistic understanding, DPP+GC perform better; similarly, for tasks that require modeling local dynamics and short-term temporal context, DPP+LC may be optimal. However, each of these fixed strategies performs well only on specific task categories and degrades noticeably on others. In contrast, HAS maintains strong performance across all evaluation settings, offering a robust and well-balanced solution that generalizes reliably to diverse video understanding scenarios.

---

### Official Review · Reviewer_zF8g · 2025-10-31

**Soundness:** 2
**Presentation:** 1
**Contribution:** 2
**Rating:** 4
**Confidence:** 4

**Summary:**

This paper proposes an adaptive, training-free frame selection method (HAS) to filter informative keyframes for Video-MLLMs. By first constructing a set of backbone frames using a determinantal point process, which is then further supplemented with local or global frames based on the text query, HAS samples the most informative frames from a long video for maximizing MLLM performance. It is intuitive, plug and play with any MLLM model, and is compared with other keyframe selection baselines (UNI / TOP / AKS) on 3 benchmarks.

**Strengths:**

- **Clear, intuitive methodology** The two-stage design (backbone frame selection + query dependent enrichment) is straightforward and easy to understand.
- **Plug-and-play / training-free** Appears to be compatible with most MLLMs, which is practical and can be used with future models as well.
- **Decent ablation suite** The paper's current ablations almost fully covers important component/budget design choices.

.

**Weaknesses:**

- **Missing critical ablation on the VL feature extractor.** The entirety of HAS depends on the quality of the features produced by the feature extractor and their measured similarities. However, no ablation or reasoning is given as to why CLIP is used and considered sufficient for this task as opposed to more recent visual-language encoders such as SigLIP or SigLIP-2.

- **Insufficient baseline coverage.** Only UNI, TOP and AKS are used as comparisons for alternative keyframe sampling methods. More recent and sophisticated methods mentioned in the Related Works, such as T* (CVPR 25), should also be compared against. Moreover, if the authors are citing papers from CVPR 25, they should also include/compare other keyframe sampling papers such as (Flexible Frame Selection for Efficient Video Reasoning). This weakens broad superiority claims.

- **Ambiguous use of SOTA** The authors often times claim achieving SOTA results (Section 4.2.1) while larger open-source models such as LLaVA-Video 72B clearly outperforms all baselines presented in the paper. which is misleading. This point leads to two questions:
	- If HAS is training-free and model agnostic, why are results with 72B models not included?
	- If simply using a larger LLM leads to best performance over all presented baselines, how impactful is keyframe selection really? LLaVA-Video 72B achieves 62.1 on LongVideoMME with 64 frames sampled uniformly; again, applying HAS here is crucial to truly measure its impact on downstream performance.

- **Lack of qualitative results**
	- The paper shows two qualitative examples in one figure but lacks a sufficient, quality investigation. Examples on the hour-long videos that HAS is quantitatively tested would provide strong support towards the authors claim of HAS' superiority in choosing better keyframes that are also relevant to the user query.  For example, per-question-type analysis or annotation-based checks that selected frames actually contain the answer more often than baselines would be valuable.
	- There are many claims throughout the paper that HAS can adaptively focus on local regions for fine-grained queries, while broadening to more sparse selections globally. No support is given for this claim besides marginal quantitative improvements, which still doesn't fully justify this claim.
- **Poor visualizations** There are only two figures, and they are not very good. Figure 1 says "Local Context" twice instead of "Global Context" in the second block. Figure 2 is low quality and difficult to understand or see what keyframes were selected without major zoom.



## Minor comments
- **Missing runtime / cost analysis.** How long does HAS take to run? If it is negligible due to the lack of learning or auxiliary networks, it should be easy to add to the paper.
- **Candidate-fps sensitivity sweep.** The pre-sampled 1 FPS candidate pool is a design choice — show results at 0.5 / 2 / 4 FPS to test robustness to short/high-motion events.

**Questions:**

- If HAS is training-free and model agnostic, why are results with 72B models not included?
- Why do larger MLLM models outperform smaller models with keyframe selection? The performance improvement of keyframe selection seems to be less impactful than model size.
- Does using a better encoder (such as SigLIP) improve HAS performance?

---

> ### Author Response · Authors · 2025-11-23
>
> ### Q1. Missing critical ablation on the VL feature extractor. The entirety of HAS depends on the quality of the features produced by the feature extractor and their measured similarities. However, no ablation or reasoning is given as to why CLIP is used and considered sufficient for this task as opposed to more recent visual-language encoders such as SigLIP or SigLIP-2.
>
> **Response:**
> Thank you for constructive feedback!
>
> We chose to use CLIP in our method because it is the same model used in AKS, ensuring consistency with prior work. Additionally, we conducted a comparison between CLIP and SigLIP2 on videomme and lvb. Our results show that SigLIP2 performs slightly better than CLIP, but the difference in performance is marginal.
>
> | **benchmark** |  **SigLip2** | **CLIP**|
> |------------|---------|---------|
> | videomme         |   68.3     |   68.1   |
> | lvb              |   62.5     |   62.2   |
>
> ---
>
> ### Q2. Insufficient baseline coverage. Only UNI, TOP and AKS are used as comparisons for alternative keyframe sampling methods. More recent and sophisticated methods mentioned in the Related Works, such as T* (CVPR 25), should also be compared against. Moreover, if the authors are citing papers from CVPR 25, they should also include/compare other keyframe sampling papers such as (Flexible Frame Selection for Efficient Video Reasoning). This weakens broad superiority claims.
>
> **Response:**
> Thank you for constructive feedback!
>
> We conducted additional experiments with T* by re-implementing the method from its algorithmic description on videomme and lvb. The results, summarized below, show that HAS consistently outperforms both AKS and T*.
>
> For the paper “Flexible Frame Selection for Efficient Video Reasoning,” although a project page is available, the official code has not been released. Consequently, we were unable to faithfully reproduce the method, and therefore do not include this comparison in our evaluation.
>
> | **benchmark** | **T*** | **AKS** | **HAS** |
> |---------------|-----------|----------|----------|
> | videomme      | 65.8      | 67.1     | 68.1     |
> | lvb           | 61.8      | 61.1     | 62.2     |
>
>
> ---
>
> ### Q3. Ambiguous use of SOTA The authors often times claim achieving SOTA results (Section 4.2.1) while larger open-source models such as LLaVA-Video 72B clearly outperforms all baselines presented in the paper. which is misleading.
>
> **Response:**
> Thank you for constructive feedback!
>
> We would like to clarify that we use this term strictly in the context of comparisons under the same model, where our method achieves the best performance among the evaluated keyframe sampling strategies. We did not intend to claim absolute state-of-the-art performance across all model sizes or video MLLMs. To avoid this ambiguity, we have revised the wording in the manuscript to make this scope explicit.
>
> ---
>
> ### Q4. If HAS is training-free and model agnostic, why are results with 72B models not included? If simply using a larger LLM leads to best performance over all presented baselines, how impactful is keyframe selection really?.
>
> **Response:**
> Thanks!
>
> In response to the question of why we did not include results with 72B-scale models, the main reason is that long-video inference with such large models demands computational resources that are orders of magnitude higher than those required for 7B models, making comprehensive experimentation practically infeasible for us.
>
> Moreover, although a 72B model with uniform sampling may outperform a 7B model with HAS on certain benchmarks, it is important to emphasize that deploying a 72B model entails substantially higher computational costs, slower inference, and much stricter hardware requirements. By contrast, applying HAS to a lightweight 7B model allows us to narrow—and in some cases even surpass—the performance gap relative to a 72B model. This enables strong video understanding on far more accessible hardware, making our approach considerably more scalable in practice. In essence, HAS helps bridge the performance gap with only a fraction of the computational burden, which makes it a compelling choice for real-world, resource-constrained scenarios.
>
> ---
>
> ### Q5. Poor visualizations There are only two figures, and they are not very good. Figure 1 says "Local Context" twice instead of "Global Context" in the second block. Figure 2 is low quality and difficult to understand or see what keyframes were selected without major zoom.
>
> **Response:**
> Thank you for constructive feedback!
>
> We have included a more comprehensive visual analysis in the appendix, which provides clearer evidence of HAS's dynamic adaptation in various settings. Additionally, we have revised the poor visualizations to better demonstrate how HAS works

---

> ### Author Response · Authors · 2025-11-23
>
> ### Q6. Missing runtime / cost analysis. How long does HAS take to run? If it is negligible due to the lack of learning or auxiliary networks, it should be easy to add to the paper.
>
> **Response:**
> Thank you for constructive feedback!
>
> We evaluated the average time required by different methods to perform key frame selection on the videomme benchmark. The results show that our method requires a similar amount of time as AKS and TOP, while being substantially faster than T* and VideoTree.
>
> TOP, AKS, and HAS exhibit similar runtime because all three methods first use a CLIP model to compute the similarity between frame and query, and then apply different frame selection strategies based on these similarity scores. In our implementation, for example, the total 5.46 s + 43 ms required by HAS consists of 5.46 s spent on computing frame–query similarities and only 43 ms on the subsequent frame selection. Since the similarity computation dominates the overall runtime, the additional complexity of the HAS selection strategy contributes only a negligible overhead, resulting in a total time cost that remains comparable to that of AKS and TOP.
>
> | **method* | **T*** | **VideoTree** | **TOP** | **AKS** | **HAS** |
> |:------------:|:------:|:-------------:|:------:|:------:|:------:|
> |  time    | 101.24s |    12.64s     | 5.46s+1ms | 5.46s+2ms | 5.46s+43m |
>
> ---
>
> ### Q7. Candidate-fps sensitivity sweep. The pre-sampled 1 FPS candidate pool is a design choice — show results at 0.5 / 2 / 4 FPS to test robustness to short/high-motion events.
>
> **Response:**
> Thank you for constructive feedback!
>
> We evaluated the model performance at various FPS settings on videomme and lvb. We observed a significant performance improvement when the FPS increased from 1 to 2, with the model's performance rising from 68.1 to 68.7. However, as the FPS decreased from 1 to 0.25, the model's performance gradually declined.
>
> This behavior reflects an inherent trade-off between information retention and computational cost. Higher FPS preserves more visual and temporal information, which generally yields better performance. Lower FPS, on the other hand, substantially reduces the computation needed for frame–query similarity calculation, but at the cost of information loss, which ultimately results in degraded performance.
>
> | **benchmark**  | 2    | 1    | 0.5  | 0.25 |
> |----------|------|------|------|------|
> | videomme | 68.7 | 68.1 | 68.0 | 67.6 |
> | lvb      | 62.3 | 62.2 | 61.3 | 61.1 |

---

> ### Comment · Reviewer_zF8g · 2025-11-26
>
> Thank you for the rebuttal and for taking the time to run additional experiments. I have a few follow-up questions and clarifications based on your responses:
>
> ---
>
> ### **Q1 - SigLIP2 vs. CLIP Results**
>
> Thank you for adding the SigLIP2 experiment. However, the numbers are somewhat surprising given the significant improvements over CLIP reported in the SigLIP and SigLIP2 papers. Why does HAS not appear to benefit from this stronger vision-language encoder? If SigLIP2 produces higher-quality features, one would expect a corresponding improvement in the frame-selection pipeline. While I understand using CLIP as a fair comparison to AKS, this can always be added as an ablation - your method itself should not be dictated by design choices in AKS.
>
> ---
>
> ### **Q2 - Clarification on the T\* Implementation**
>
> I appreciate the effort in running the T* comparison. However I have a small question: did the authors use the official T* implementation or was the method re-implemented?
>
> If it was re-implemented, could the authors clarify why this was feasible for T* but not for the Flexible Frame Selection?
>
> ---
>
> ### **Q3 / Q4 - Clarification on “Lightweight” Claims and the Impact of LLMs**
>
> This is still one of my core concerns. While 7B models are indeed smaller than 72B models, they are still quite large, and it is difficult to categorize them as *lightweight* or as part of an *efficient scaling* paradigm. This impacts the stated motivation of HAS as a practical and broadly applicable method, especially given the fact that previous works such as T* do include these comparisons. The fact that simply larger LLMs with uniform sampling outperforms HAS was not fully addressed in the author's rebuttal.
>
> On this note, I would recommend avoiding the term SOTA in the context of “SOTA performance on long-video benchmarks”, as SOTA conventionally refers to *across all models*, not only among a subset of baselines. The authors still incorrectly claim SOTA results (Lines 356, 404) - there are other ways to distinguish that HAS outperformed other baselines than claiming SOTA performance.
>
> ---
>
> ### **Q5 - Visualization Quality and Demonstration of HAS Benefits**
>
> I reviewed the appendix visualizations, but they still only show examples of the selected frames from HAS, rather than demonstrating cases where HAS provides a clear advantage over UNI, TOP, or AKS. For a method whose claim is adaptive, query-conditioned sampling, it would be helpful to include qualitative examples where:
>
> - other baselines fail to capture the relevant frames for the query,  and
> - HAS successfully identifies them.
>
> In Figure 2, only 8 frames are sampled - it would be better to visually compare with UNI, TOP, and AKS at 16, 32, and 64 frames. I am still not quite convinced on HAS' ability to sample query-dependent frames better than previous methods, especially given the very small quantitative improvements over AKS in Tables 2 and 3 on most datasets. This is also a strong concern of mine regarding the contribution of the work and supporting the claims in the paper, especially given the marginal (<1-2%) improvements of HAS over AKS on LVB and VideoMME.
>
>
> ---
>
> ### **Q6 / Q7**
> These have been fully addressed, thank you.

---

> > ### Author Response · Authors · 2025-11-27
> >
> > ### Q1 - SigLIP2 vs. CLIP Results
> >
> > **Response to Q1 – SigLIP2 vs. CLIP Results**
> >
> > Thank you for the constructive feedback.
> >
> > We have added an ablation study over different vision encoders (CLIP vs. SigLIP2) in the appendix to clarify their impact on HAS.
> >
> > As for why SigLIP2 yields only small improvements in our experiments, our view is twofold:
> >
> > 1. **Selecting more relevant frames is not the sole factor governing performance.**
> >    Even though SigLIP2 is a stronger image–text encoder than CLIP, HAS’s final performance is not determined by keyframe quality alone. The frame set passed to the VLM is a combination of (1) backbone frames, (2) global context frames, and (3) local context frames. Keyframes are important, but they are only one component of the overall contextual evidence. For many questions, such as “What is this article/video mainly about?” or “What does the person do before they start X?”, the answer depends more on sufficient global or short-range temporal context than on the query-relevant frame. In these cases, making the keyframes slightly more accurate (via SigLIP2) may not change the downstream QA performance significantly, because the bottleneck lies in whether the necessary context is present in the combined frame set, rather than in the precision of the top similarity-ranked frames.
> >
> > 2. **Benchmark characteristics and VLM capacity limits diminish the marginal effect of a stronger encoder.**
> >    In our experiments across multiple long-video benchmarks, we observe that a non-trivial portion of questions cannot be reliably solved from visual frames alone: they often require audio cues or external knowledge that neither CLIP nor SigLIP2 provides. Moreover, the VLM has an intrinsic capacity limit—better frame sampling can surface evidence that helps the model approach this limit, but it cannot increase the limit itself. Practically, we see cases where, even when we supply frames that are clearly “sufficient” from a human perspective, the VLM still fails to answer correctly. For these questions, the limiting factor is the VLM’s inherent capability rather than the quality of the key frame, so simply replacing CLIP with SigLIP2 does not meaningfully change the outcome.
> >
> > Taken together, we believe these two factors can partially explain why SigLIP2 does not lead to substantial performance improvements in our setting.
> >
> > ---
> >
> > ### Q2 - Clarification on the T* Implementation
> >
> > **Response:**
> >
> > Thanks!
> >
> > We confirm that we used the official T* implementation released by the authors, rather than re-implementing the method. Concretely, we followed the default configuration in the public codebase, including using GPT-4o as the grounder and OWL-ViT as the heuristic model, and we kept all other hyperparameters and settings unchanged. The only modification we made was to set the maximum number of searched frames to 64, so that T* operates under the same frame budget as our method for a fair comparison.
> >
> > ---
> >
> > ### Q3 / Q4 - Clarification on “Lightweight” Claims and the Impact of LLMs
> >
> > **Response:**
> >
> > Thank you for constructive feedback!
> >
> > To further assess how HAS scales to larger vision–language models, we additionally evaluated HAS with the InternVL3-38B backbone and with GPT5 on the VideoMME benchmark. Due to computational constraints, for GPT5 we evaluated only 128-sample subsets of the VideoMME-medium and VideoMME-long splits, obtained by uniformly sampling 128 videos from each split. For InternVL3-38B, we followed the same frame budget and evaluation protocol as in the 7B setting and evaluated on the medium and long splits.
> >
> >
> > | Backbone           | Sampling   | medium | long |
> > |--------------------|------------|----------------|--------------|
> > | InternVL3-38B      | UNI        |  72.1          |  62.1      |
> > | InternVL3-38B      | HAS        |  73.8           | 64.0       |
> >
> > | Backbone           | Sampling   | medium(subset) | long(subset) |
> > |--------------------|------------|----------------|--------------|
> > | GPT5      | UNI        | 77.3          | 75.9       |
> > | GPT5      | HAS        | 81.3          | 78.1       |
> >
> > As shown, HAS consistently outperforms uniform sampling on both backbones, indicating that HAS remains effective and delivers clear improvements as model capacity increases.
> >
> > We also evaluated HAS with more lightweight backbones. Under the same frame budget and experimental setup, HAS consistently improves model capacity:
> >
> > | Backbone | Sampling   | short | medium | long |
> > |-----------------|------------|-------|-------|-------|
> > | InternVL3-1B          | UNI                     | 61.9  | 46.8 | 39.2|
> > | InternVL3-1B          | HAS                     | 62.4  | 49.8 | 41.6|
> >
> > For term "SOTA", We have revised all related descriptions in the revisied manuscript and removed the use of the term “SOTA” to avoid any potentially confusing or inaccurate wording.

---

> > > ### Author Response · Authors · 2025-11-27
> > >
> > > ### Q5 - Visualization Quality and Demonstration of HAS Benefits
> > >
> > > **Response.**
> > >
> > > Thank you for constructive feedback!
> > >
> > > In the revisied manuscript, we have substantially expanded the qualitative analysis in the appendix and now include visualizations comparing HAS with UNI, TOP, and AKS under different frame budgets.

---

### Author Response · Authors · 2025-11-23

Firstly, sincerely thank all the reviewers for their efforts in reviewing our paper and providing constructive suggestions. We are greatly encouraged
that the reviewers find that

- Our idea is interesting and well-motivated. (Reviewer 6Z8Q)
- The proposed method consistently outperforms prior frame-sampling methods across three benchmarks. (Reviewer Hx4z and CRZf)
- The paper is clearly written. (Reviewer Hx4z and 6Z8Q)
- Ablation studies are thorough. (Reviewer zF8g and CRZf)

Secondly, as for the concerns and suggestions raised by each reviewer, we have done our best to address them thoroughly and have provided
detailed responses to each of them. In the following, we summarize our responses to the main concerns raised by the reviewers.

- Reviewer zF8g： We added ablations on the VL encoder and expanded the baseline comparisons to strengthen the empirical evaluation. We clarified the intended meaning of “SOTA” and our focus on 7B models rather than 72B models in terms of computational cost and practicality. We also provide additional qualitative visualizations, as well as runtime and FPS-sensitivity analyses.

- Reviewer 6Z8Q: We clarified the distinctions between HAS and prior hierarchical long-video methods and added experiments on longer and more dynamic videos understanding benchmarks. We further included a runtime analysis and clarified the inconsistency in performance improvements in ablation study.

- Reviewer Hx4z: We clarified the role of DPP versus TOP and its connection to our hierarchical design, explained how the frame budget is adaptively allocated between DPP-selected backbone frames and contextual frames and provided runtime analyse.

- Reviewer CRZf: We revised the manuscript to clarify the limitations of previous methods, explain how HAS addresses these issues, and improve the visualization in Fig. 2. In addition, we investigated the effect of the threshold $\tau$ in Adaptive Contextual Enrichment and add experiments on temporal grounding tasks.

In the revised manuscript, we have refined some visualizations ambiguous descriptions. The changes have been highlighted in blue.

---

### Author Response · Authors · 2025-12-01

We thank the reviewers for their constructive feedback. Below, we summarize our responses to the reviewers’ comments, including the clarifications on specific issues, revisions made to the manuscript, and the additional experiments conducted.

## Clarifications on Specific Issues

1. **Comparison with other frame sampling and hierarchical methods.** We clarify that HAS is, to our knowledge, the first frame sampling method to use DPP to jointly enforce query relevance and low redundancy, and to infer the query’s information needs to adaptively allocate the remaining frame budget between local and global context. While VideoTree and HierarQ also adopt hierarchical or global/local design ideas, their concrete mechanisms and goals differ substantially from HAS.

2. **Aggressive 1 FPS pre-sampling.** We clarify that the choice of 1 FPS is a practical configuration rather than an inherent limitation of HAS. It aims to balance the computational cost with model performance. For videos with rapid scene changes, we can straightforwardly adopt a higher pre-sampling FPS.

3. **Limited performance gains of DPP compared to TOP.** We clarify that, due to its non-redundant selection, DPP excels on benchmarks requiring broad semantic coverage across the video, but is only slightly better than TOP when most questions are answerable from a single scene. This motivates our local context branch in the context enrichment stage, which explicitly recovers short-term temporal dynamics around DPP-selected keyframes.

---

## Revisions to the Manuscript

Guided by the reviewers’ suggestions, we have made the following revisions to the manuscript.

1. **Revised related work.** In the related work section, we provide a more precise discussion of the limitations of prior video frame sampling methods and explicitly highlight the advantages of AKS.

2. **Corrected inaccurate descriptions.** In the original version, we used the term “SOTA” to indicate that, under the same backbone, our sampling strategy achieves the best performance among compared sampling methods. Following the reviewers’ suggestions, we revise this to avoid ambiguity.

3. **Improved qualitative results and added more examples.** We updated the qualitative results in the main paper to provide clearer and more representative cases, and in the appendix we provide substantially more qualitative results under different frame-sampling budgets.

---

## Additional Experiments

To address the reviewers’ concerns, we have provided substantial experimental evidence.

1. **Runtime analysis.** We evaluate the time cost of different frame sampling methods under the same setting: HAS has an average time cost of 5,503 ms, which is close to TOP (5,461 ms) and AKS (5,462 ms), and much faster than T* (101,240 ms) and VideoTree (12,640 ms).

2. **Scaling to different model sizes.** We further evaluate HAS across backbones of different sizes: on InternVL3-38B, VideoMME-Long improves from 62.1 to 64.0; on GPT5, performance on the VideoMME-Long subset improves from 75.9 to 78.1; and on InternVL3-1B, VideoMME-Long improves from 39.2 to 41.6, showing consistent gains across backbone scales.

3. **Additional video VQA benchmarks.** We further evaluate HAS on several other video VQA benchmarks: on EgoSchema, compared with UNI, accuracy improves from 71.4 to 74.4; on MMBench, the score improves from 1.70 to 1.96; on MovieChat, accuracy improves from 40.5 to 42.9, indicating that HAS can generalize well across different video VQA benchmarks.

4. **Hyperparameter τ ablation.** We perform a sensitivity analysis on the hyperparameter τ and find that in the range τ ∈ [0.5, 0.9] the method is largely insensitive and maintains stable performance, while in the range τ ∈ [0.1, 0.4] performance degrades to some extent, showing that HAS is robust within a fairly broad and practically reasonable τ regime.

5. **Visual grounding QA.** On the visual grounding QA benchmark ReXTime, we evaluate HAS and obtain improvements from 30.2 to 31.9 in mIoU, and from 21.2 to 23.7 in Acc@IoU > 0.5, indicating that HAS not only helps VQA tasks but also improves visual grounding ability.

6. **Different visual encoders.** We study the impact of using a stronger visual encoder by replacing CLIP with SigLIP2; this yields modest but consistent gains across benchmarks (e.g., from 68.1 to 68.3 on VideoMME and from 62.2 to 62.5 on LVB).

7. **Comparison with other baseline.** We further compare HAS with T* (CVPR 2025) on VideoMME and LVB: on VideoMME, HAS achieves 68.1 compared to 65.8 for T* , and on LVB, HAS obtains 62.2 versus 61.8 for T*, highlighting effectiveness for long video reasoning.

In summary, we believe that the above clarifications, revisions, and additional experiments adequately address the reviewers’ concerns and further strengthen the technical soundness and empirical support of our work. We hope that these updates will help convey the contribution and significance of HAS more clearly

---

### Meta-Review · Area_Chair_3Cx4 · 2026-01-05

**Summary:**

After reading the paper and discussion, the AC agrees with the concern raised by reviewers, including insufficient baseline coverage (zF8g), the absence of large MLLM results (zF8g), limited novelty (6Z8Q) and improvement (Hx4z). Based on this, the AC tends to reject this paper.

**Reviewer Concerns:**

Most concerns are answered by the authors. However, the author failed to provide sufficient results for larger MLLMs (zF8g), and the discussion on novelty and improvement is somewhat limited.

**Reviewer Scores:**

Reviewer zF8g challenged the lack of baseline and larger MLLM reuslts; however, the authors failed to provide sufficient results. So the score may not change
Reviewer 6Z8Q mainly challenged the limited novelty and pipeline design. The authors' discussion is a bit weak on this point. So the score may not change
Reviewer Hx4z raised the score from 4 to 6
Reviewer CRZf did not reply to the rebuttal, but the initial score is positive.

---

### Decision · Program_Chairs · 2026-01-26

Reject